# Classical complement pathway inhibition reduces brain damage in a hypoxic ischemic encephalopathy animal model

**Parvathi Kumar**[1,2]*, **Pamela Hair**[1], **Kenji Cunnion**[1,2,3], **Neel Krishna**[1,2,3], **Thomas Bass**[2,3]

**1** ReAlta Life Sciences, Norfolk, VA, United States of America, **2** Department of Pediatrics, Children's Hospital of The King's Daughters, Norfolk, VA, United States of America, **3** Eastern Virginia Medical School, Norfolk, VA, United States of America

* pkumar@realtals.com

**Data Availability Statement:** All relevant data are within the manuscript and its Supporting Information files.

**Funding:** The funder provided support in the form of salaries for authors [PK,PH,KC, NK], but did not

## Abstract

Perinatal hypoxic ischemic encephalopathy (HIE) remains a major contributor of infant death and long-term disability worldwide. The role played by the complement system in this ischemia-reperfusion injury remains poorly understood. In order to better understand the role of complement activation and other modifiable mechanisms of injury in HIE, we tested the dual-targeting anti-inflammatory peptide, RLS-0071 in an animal model of HIE. Using the well-established HIE rat pup model we measured the effects of RLS-0071 during the acute stages of the brain injury and on long-term neurocognitive outcomes. Rat pups subject to hypoxia-ischemia insult received one of 4 interventions including normothermia, hypothermia and RLS-0071 with and without hypothermia. We measured histopathological effects, brain C1q levels and neuroimaging at day 1 and 21 after the injury. A subset of animals was followed into adolescence and evaluated for neurocognitive function. On histological evaluation, RLS-0071 showed neuronal protection in combination with hypothermia (P = 0.048) in addition to reducing C1q levels in the brain at 1hr (P = 0.01) and at 8 hr in combination with hypothermia (P = 0.005). MRI neuroimaging demonstrated that RLS-0071 in combination with hypothermia reduced lesion volume at 24 hours (P<0.05) as well as decreased T2 signal at day 21 in combination with hypothermia (P<0.01). RLS-0071 alone or in combination with hypothermia improved both short-term and long-term memory. These findings suggest that modulation by RLS-0071 can potentially decrease brain damage resulting from HIE.

## Introduction

Perinatal asphyxia with moderate to severe hypoxic ischemic encephalopathy (HIE) is a significant public health concern worldwide with varying incidence in high-, middle- and low-income countries. Approximately 1-2/1000 live-births are affected in the developed world with an eight-fold higher incidence in the under developed countries [1]. Without adequate management, 10–60% of affected infants die worldwide and at least 25% of survivors are affected with significant neurodevelopmental disabilities [2]. Neuroprotective therapy using whole

have any additional role in the study design, data collection and analysis, decision to publish, or preparation of the manuscript. The specific roles of these authors are articulated in the 'author contributions' section.

**Competing interests:** Commercial affiliation with ReAlta Life Sciences Inc, does not alter our adherence to PLOS ONE policies on sharing data and materials.

body cooling also referred to as therapeutic hypothermia (TH) has become the standard of care management for HIE. TH improves survival in neonates with HIE [3], but offers only a modest reduction in risk of death or severe disability, from 53% to 45% [4]. Among term neonates with moderate or severe HIE, longer or deeper cooling, or both, does not reduce death, or moderate to severe disability at 18 months of age and may worsen mortality [5].TH is associated with undesirable side effects, requires specialized equipment with treatment in tertiary care centers and is effective only if initiated within 6 hours of birth [2,6]. Despite this vital unmet medical need, no pharmacological adjunct or alternative therapy has proven beneficial improving outcomes in neonatal HIE.

Recent studies have demonstrated an important role for complement-activated inflammation in ischemia-reperfusion injury associated with HIE [7–9] as well as MPO and oxidative stress, in contributing to brain tissue damage [10]. Following ischemia-reperfusion, neoantigens are expressed on injured vascular endothelial cells that bind C1 [11] leading to the expression of C1q [12]. Further evidence of HIE induced activation of the classical complement pathway is the finding of C5a, a potent pro-inflammatory mediator, in the asphyxiated brain [13], with reduced brain infarction, less neurofunctional deficits, and attenuated mitochondrial reactive oxygen species in C1q deficient mice subjected to hypoxic-ischemic brain injury [8,9]. The deposition of C1q, C3, C3 split products, and C9 have been associated with greater extent of brain injury in animal models [8,14].

Therapeutic hypothermia modulates complement activation in a complex manner [8,14]. A recent study by Shah et al reported that therapeutic hypothermia was associated with reduced brain infarction, decreased mitochondrial expression of C1q, decreased microglial and neuronal deposition of C3 and C9, and reduced systemic levels of C1q and C5a [15].

New evidence has begun to elucidate the role of neutrophils in HIE demonstrating rapid recruitment and the generation of neutrophil extracellular traps (NETs) [16]. Microglia are considered the "neutrophils of the brain" containing MPO and the ability to generate extracellular traps [17,18]. In HIE animal models, microglia are activated into a pro-inflammatory state and are believed to be major contributors to the inflammatory brain damage during reperfusion [19,20].

RLS-0071, also known as Peptide Inhibitor of Complement C1 (PIC1), is a 15 amino acid peptide that binds to C1, inhibiting the enzymatic activity of the serine protease tetramer, C1s-C1r-C1r-C1s and activation of the classical complement pathway [21,22]. RLS-0071 is also an antioxidant that inhibits single electron transport and hydrogen atom transport [23,24], inhibits myeloperoxidase activity and blocks neutrophil extracellular trap (NETosis) formation [23]. Originally RLS-0071 was identified as a complement inhibitor and over the course of the last 24 months our understanding of RLS-0071 has evolved to reveal that it is a peptide with additional anti-inflammatory mechanisms of action that go far beyond the complement system. Here we explore the impact of RLS-0071 on the disease process in an animal model of neonatal HIE and the extent to which the RLS-0071 enhanced neuroprotection by limiting brain damage.

## Materials and methods

### Ethics statement

All animal experiments were performed under an approved protocol by the Eastern Virginia Medical School (EVMS) Institutional Animal Care and Use Committee. Where applicable, all animal experiments were carried out according to the National Institute of Health (NIH) guidelines for the care and use of laboratory animals and approved by the National Ethics Committee (National Animal Experiment Board, Finland). The animal facility was accredited

by the Association for Assessment and Accreditation of Laboratory Animal Care (AAALAC), International.

## Materials

RLS-0071 (IALILEPICCQERAA-dPEG24) was manufactured by PolyPeptide Group (San Diego, CA) to ≥ 95% purity as verified by HPLC and mass spectrometry analysis. RLS-0071 was solubilized in a 0.05 M Histidine buffer at pH 6.7 (normal saline with 0.01 M $Na_2HPO_4$ buffer to 37.5 mM). Primary antibodies used for assays included mouse anti-rat C1q (Abcam), goat anti-human C1q (A200,Complement Technology, Inc., Tyler, Texas) and secondary antibodies included goat anti-mouse horseradish peroxidase (HRP) (A4416, Sigma-Aldrich), and donkey anti-goat IgG (H+L) Alexa Fluor (AF) 488 (A-11055, Life Technologies, Grand Island, NY).

## Animal model of unilateral hypoxia-ischemia

Pregnant Wistar rats at embryonic day 19 (Hilltop Lab Animals Inc., Scottsdale, PA) were housed individually and allowed to spontaneously deliver in-house. Controlling for littler effect, the pups were randomized on the day of birth (10/litter). When considered 'term equivalent' at P10, hypoxic injury was induced using the Vannucci method of carotid ligation [14] after the pups (including both males and females) were randomly allocated to the following groups: normothermia (NT), hypothermia (HT), RLS-0071 given as treatment starting at 60 minutes after hypoxic injury maintained at normothermia (RLS-0071) and RLS-0071 given at the same schedule as above with hypothermia (HT+ RLS-0071) (Fig 1). As described by Shah et al., the animals were subjected to right sided carotid ligation to cause unilateral ischemic injury [14]. After recovery at 37±1˚C over 60 minutes, the pups were subjected to hypoxia at 8% $O_2$/balance nitrogen for 45 minutes at 37˚sC. Pups in the HT and HT+ RLS-0071 groups were maintained at target rectal temperatures of 31–32˚C for 6 hours by placing them in a temperature-controlled chamber set to 28–30˚C. Pups in the NT and RLS-0071 groups were kept in a different chamber at 37±1˚C. When RLS-0071 was administered, it was administered as 2 doses of 10mg/kg, intraperitoneal (i.p.) given 4 hours apart starting 60 minutes after removal from the hypoxia chamber. The second dose of RLS-0071 was administered while the animals were in the hypothermia chamber, 4 hours after the first dose. After each intervention, the pups were rewarmed to 37±1˚C on a homeothermic blanket system. Each animal used varying times to rewarm but all of them rewarmed in 60 minutes. When all the animals had rewarmed, they were returned to the nursing dam en masse. Rats were euthanized at different time points with a lethal dose of pentobarbital (FatalPlus™). Brain examination for gross infarction and histology was performed after euthanasia at 48 hours post-hypoxia. Cohorts for neurocognitive testing were allowed to grow to young adult age and then were tested for long-term spatial memory retention, novel object recognition, and locomotive function.

For the neuroimaging studies, the rat pups received 2 doses of RLS-0071 at 0.25mg/rat (approximately 12.5mg/kg) subcutaneously (s.c.) starting 1 hour after removal from the hypoxia chamber, given 4 hours apart. Additionally, there was a control group included (Naïve). This group of animals did not undergo any surgical procedure or hypoxic insult and were maintained at normothermia as described above.

## Blood collection, tissue harvest and processing

For histological evaluation the euthanized animals were perfused with ice-cold PBS. The right and left hemisphere of the brains were harvested and stored in liquid nitrogen until needed. When brains were collected for histopathology, after perfusion with ice cold PBS, additional perfusion using 10% neutral buffered formalin (NBF) was performed. The brains were then

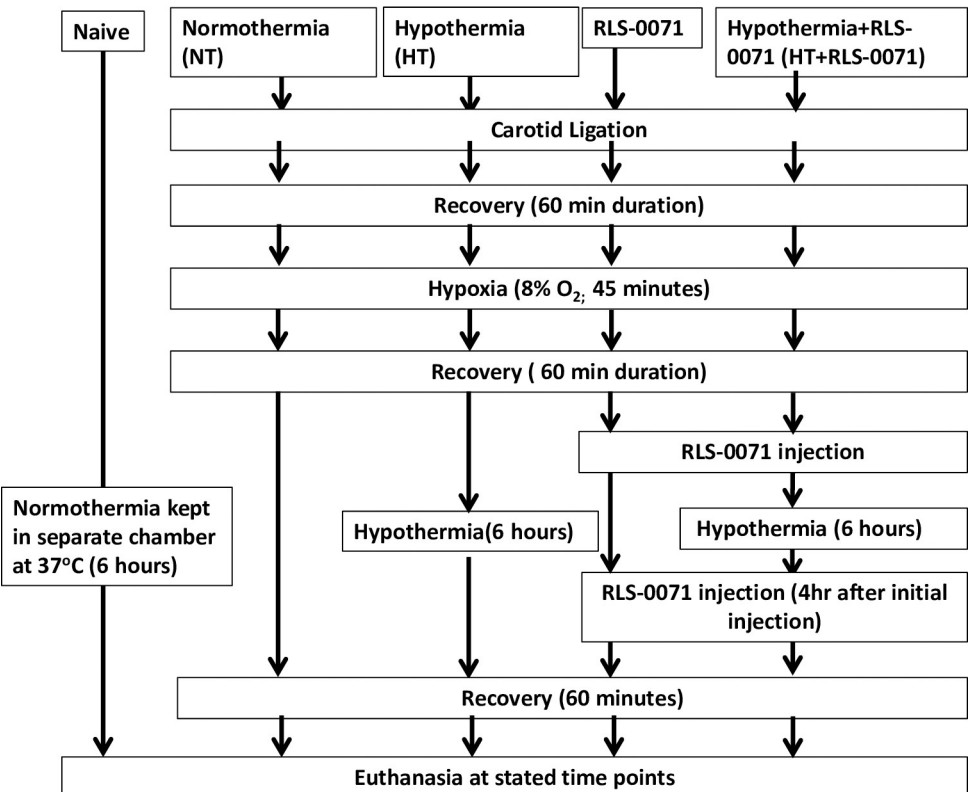

**Fig 1. Procedural flow for study groups.** Term equivalent rat pups at P10 were randomly assigned to 4 different groups: Normothermia (NT), hypothermia (HT), treatment with RLS-0071 only (RLS-0071) and treatment with hypothermia and RLS-0071 (HT+RLS-0071). Experimental animals underwent unilateral carotid ligation followed by exposure to hypoxia (8% $O_2$/balanced nitrogen, Vannucci model) for 45 minutes. HT animals and HT+RLS-0071 animals were placed in open jars in a temperature-controlled chamber to maintain a target rectal temperature of 31–32°C for 6 h, NT, and RLS-0071 animals were kept in a separate chamber at 37±1°C. RLS-0071 animals were injected with 2 doses of RLS-0071, starting 60 minutes after removal from the hypoxia chamber, with a second dose given four hours later. After intervention, pups were rewarmed, placed back with the dam, and euthanized at stated time points. For the neuroimaging studies a control group was included (Naïve). This group of animals did not undergo any surgical procedure or hypoxic insult and were maintained at normothermia.

paraffin embedded (Excalibur Pathology, Inc., Norman, OK) after processing to acquire 5 μm coronal sections using a RM2125 rotary microtome (Leica Microsystems) for histopathology purposes.

Right hemispheres of brains previously perfused and harvested were thawed on ice. 1 nM/mL PMSF and Pierce inhibitor tablets (Thermo Fisher, Waltham, Ma) (1 per 10mL) were added fresh to cold homogenization buffer (1% Triton X-100, 0.05M Tris-HCL, 0.15M NaCl, pH 7.0–7.5). 500μl of buffer was added to a 2 ml ceramic bead tube (Fisher Scientific), to which the entire hemisphere was added. The tubes were agitated on a Bead Mill 4 (Fisher Scientific) at 5 rpm for 120 seconds and then the beads were sedimented. The supernatants were transferred to a new microfuge tube and sedimented at 6,700 rpm for 10 minutes at 4°C. The supernatant was recovered and frozen at -20°C.

## Histopathology

Cresyl Violet staining was performed using previously described protocols [14,25] to demonstrate Nissl bodies of neurons as a measure of neuronal density in brain tissue. A digital camera

(DP70, Olympus Center) mounted on a BX50, Olympus microscope was used to capture images, and Image J (National Institutes of Health) was used for analysis.

## ELISA

To measure C1q levels in the brain, 150 μg of brain lysate in 1×PBS was added to Corning® 96-well black flat bottom polystyrene high bind microplate (Steuben County, NY) and incubated overnight at 4˚C. The plates were blocked with 10% normal donkey serum (NDS) for 1 hour after being washed with 1x PBS-0.1% Tween 20 (PBST) three times. Goat anti-human C1q (Complement Technology, Inc., Tyler, TX) at 1:50 in NDS was added to the wells and incubated at RT for 1 hour. After washing, donkey anti-goat Alexa Fluor (AF) 488 (Life Technologies, Grand Island, NY) at 1:500 dilution in NDS was added followed by a 1 hour incubation at RT. The plate was washed and blocked with 10% normal goat serum (NGS) at RT over 1 hour. The wells were washed, and the plate read using a Synergy HT (BioTek, Winooski, VT).

## T2-MRI for lesion and edema

The HIE MRI studies were performed by Charles River Discovery Research Services Finland–study number C174319. MRI acquisitions were performed at 24h after surgical procedure to confirm the development of a lesion. Additional, T2-volumetric MRI was performed 21 days after surgery to measure infarct volume and edema in all rats. MRI was performed in a horizontal 7.0 T magnet with bore size 160 mm equipped with a gradient set capable of maximum gradient strength 750 mT/m and interfaced to a Bruker Avance III console (Bruker Biospin GmbH, Ettlingen, Germany). A volume coil (Bruker Biospin GmbH, Ettlingen, Germany) was used for transmission and a two-element surface array coil was used for receiving (Rapid Biomedical GmbH, Rimpar, Germany). Isoflurane-anesthetized rats (70% $N_2O$ and 30% $O_2$; flow 300 ml/min, induction with 5%, maintenance 1.5%) were fixed to a head holder and positioned in the magnet bore in a standard orientation relative to gradient coils.

For the determination of lesion and edema volumes, absolute T2 maps were acquired with multi-slice multi-echo sequence with following parameters; (repetition time) TR = 2500 ms, (echo time) TE = 10–120 ms in 10 ms steps, matrix of 256x128, (Field-of-View) FOV of 20x20 mm2, 4 transitions and 18 coronal slices of thickness 0.7 mm. For the evaluation of edema, volumes of contralateral hemisphere, ipsilateral healthy tissue and lesion were determined, and the edema volume was given as subtraction of contralateral volume from ipsilateral hemisphere. Manual region of interest analysis for volumes was performed using in-house written Matlab software (MathWorks Inc., Natick, MA) with observer blinded to the treatment groups. T2-relaxation times (in milliseconds) give rise to the lesion contrast and are descriptive for tissue water environment in regards of local field dephasing effects spins experience in the tissue. In intact, healthy tissues, the dephasing effects are larger (shorter T2-values) than in lesioned tissues where the net-water is increased (or water re-distributed between the water compartments). Control values for absolute T2 were extracted from contralateral cortex.

## Neurobehavioral assays

**Barnes maze.** The Barnes maze was used to measure the acquisition and retrieval of a long-term spatial memory task 6 weeks after HIE. The Plexiglas maze was 122 cm in diameter, with 20 equidistant holes (10.5 cm diameter), spaced every 12˚, and centered 7.5 cm from the outer perimeter. An escape box (8cm in depth) was placed under one of the holes. The position of the escape box was varied randomly from rat to rat but kept constant for a given rat throughout testing. A floodlight was placed over the maze to serve as an aversive stimulus.

Prior to testing, rats underwent habituation sessions once a day for 5 consecutive days, during which they were placed in a holding box in the middle of the maze and allowed to acclimate for 60 sec. The holding box was lifted off the rats and the rats were allowed to explore. The test was over when either the rat found the burrowing hole and burrowed for 10 sec or the 5 min time limit was reached. If animals failed to reach the escape hole by the 5 min mark, they were led to the escape hole and allowed to burrow inside for 30 sec. On the 14th day after the habituation sessions, the rats were tested to evaluate memory retention. Latency to investigate any hole, escape latency, and number of errors during the test trials were analyzed by two-way ANOVA, followed by Bonferroni's post hoc comparison matrix. The same measurements recorded during the memory retention test were analyzed by Student's t-test. Significance was considered at $p \leq 0.05$.

**Novel object recognition.** The novel object recognition test (NOR) was performed at 8 weeks after HIE to test for long-term object memory, a cognitive function by evaluating the differences in exploration time of novel and familiar objects [26]. Rats were trained for 2 days prior to testing. During the training sessions, the rat was allowed to explore the testing chamber, a polycarbonate box ($40 \times 40$ cm) with 2 identical objects for 3 minutes. On testing day, one of the familiar objects was replaced with a novel object, made with the same material but a different shape, and the rat allowed to explore for 3 minutes. Video recordings of the interactions were reviewed by blinded investigators. Object exploration was defined as the rat directing its nose toward the object at a distance of less than 2 cm. The relative percentage of time spent with novel vs. familiar object was calculated. Interaction was determined to be sniffing, looking, or climbing on the object or the object platform. If the rat was touching the object but looking elsewhere, that time was not counted as interaction. To control for bias the rats were divided into two groups, each having a different novel and familiar object from the other group.

**RotaRod task.** A RotaRod (Harvard Apparatus, Holliston, MA) was used to test sensorimotor coordination [27]. Training and testing were completed on the same day. During training, the RotaRod rotated at 4 rpm. Rats were placed on the moving rod for 5 minutes. If the rat fell, it was placed back onto the rod. Rats rested for one hour after training. During testing the rats were placed back on the RotaRod and the rotation speed was steadily increased from 4 rpm to 40 rpm over 5 minutes. The time the rat was able to stay on the RotaRod before falling was measured.

## Statistical analysis

Means and standard error of the means (SEMs) were calculated from independent experiments. Statistical comparisons were made using the paired t-test and ANOVA where appropriate.

Medians for behavioral data are compared using notched box-plots (Box represents the interquartile range (25-75th percentile); Horizontal line represents the median; Notch represents the confidence interval; If two boxes' notches do not overlap, there is 95% confidence their medians differ) [28] Statistical analysis was performed with OpenEpi (Emory University) and SAS V9.3 (Cary, NC).

## Results

### Lowest effective treatment dose of RLS-0071 for HIE

We started by determining the lowest effective dose of RLS-0071 given as rescue treatment, one hour after removal from the hypoxia chamber. The pups were euthanized at 48 hours after hypoxia. The fresh brains were grossly visualized to evaluate for white, unperfused areas

indicative of infarction (S1A Fig). In this experiment we tested rescue doses with the following groups: (1) RLS-0071 given at 1 mg/kg × 2 doses, (2) RLS-0071 given at 5 mg/kg × 2 doses, (3) RLS-0071 given at 10 mg/kg × 2 doses, and a control group (4) normothermia with no RLS-0071 (Naïve). The NT and 1mg/kg x 2 RLS-0071 group rat pups both showed gross infarction in 100% of the pups (S1B Fig). The group receiving RLS-0071 at 5mg/kg x 2 showed gross infarction in 3 of 4 pups (75%) and the group receiving RLS-0071 at 10 mg/kg × 2 group showed gross infarction in 2 of 4 pups (50%). These data show a dose response suggesting that dosing with RLS-0071 at 10 mg/kg × 2 is the least effective dose at modulating HIE related gross findings in this animal model. Additional experiments testing doses higher than 10mg/kg × 2 demonstrated no additional benefit (data not shown).

## RLS-0071 + hypothermia provides additive effect for neuronal protection in HIE

Fig 1 shows the procedural flow for the study groups. All groups received hypoxia and unilateral carotid ligation and consisted of: (1) normothermia (NT) or no treatment, (2) hypothermia (HT), (3) RLS-0071 given at 10 mg/kg × 2 doses without hypothermia (RLS-0071), and (4) RLS-0071 given at 10 mg/kg × 2 with hypothermia (HT+ RLS-0071). 'RLS-0071+ HT' animals were immediately placed in the hypothermia chamber after receiving RLS-0071. We evaluated whether dosing with RLS-0071 at 10 mg/kg ×2 prevented neuronal loss in the affected cortical hemisphere as assessed by Cresyl violet staining. Cresyl violet is a histological stain taken up by the Nissl bodies of neurons as a measure of neuronal density in brain tissue. Dense indigo staining occurs in intact brain structures with a high density of neurons. Cresyl violet stained brain tissue was evaluated under three levels of magnification. Under 2× magnification (Fig 2A), Nissl staining for the NT control group showed variable staining with areas of low stain uptake indicative of patchy regions with marked neuronal loss. Integrated density measurements performed for each group at 2× magnification showed a progression of neuronal preservation (Fig 2D) where the RLS-0071+HT group demonstrated maximal protection with a 22% increase in neuronal density compared with the HT group (P = 0.048). At 10× magnification (Fig 2B) representative images showed increased neuron staining in the outer cortex for the RLS-0071 and RLS-0071+HT groups compared to the NT group. At 20× magnification (Fig 2C) the NT group showed few intact neurons compared with the treatment groups (animals receiving RLS-0071 or HT or HT + RLS-0071). These studies show that treatment with RLS-0071 at 10 mg/kg ×2 in combination with hypothermia yielded optimal neuronal preservation in the cortex of animals subject to HIE compared with hypothermia alone or no treatment. This suggests that RLS-0071 in combination with hypothermia yields an additive effect for neuronal preservation.

## RLS-0071 and HT + RLS-0071 decreased brain C1q deposition in HIE

Previous human and animal model data suggests that C1/C1q play an important role in mediating ischemia reperfusion injury and brain damage in HIE [7,8,13]. In order to assess whether RLS-0071, or hypothermia, alone or in combination alter C1q levels in the brain, protein preparations of brain hemispheres from the side subjected to carotid ligation were measured for C1q by ELISA for the following groups: (1) NT, 2) HT, 3) RLS-0071, and 4) HT+RLS-0071. Animals were euthanized at 1 hour or 8 hours after RLS-0071 administration (2 hours or 9 hours after hypoxia), or equivalent timepoints for the other groups, and a terminal blood draw was performed. At the 1-hour timepoint, the HT group showed no change in C1q in the brain tissue compared with NT (S2 Fig). Animals receiving RLS-0071 alone demonstrated decreased C1q in the brain tissue compared with HT (P = 0.01) and the group of animals receiving

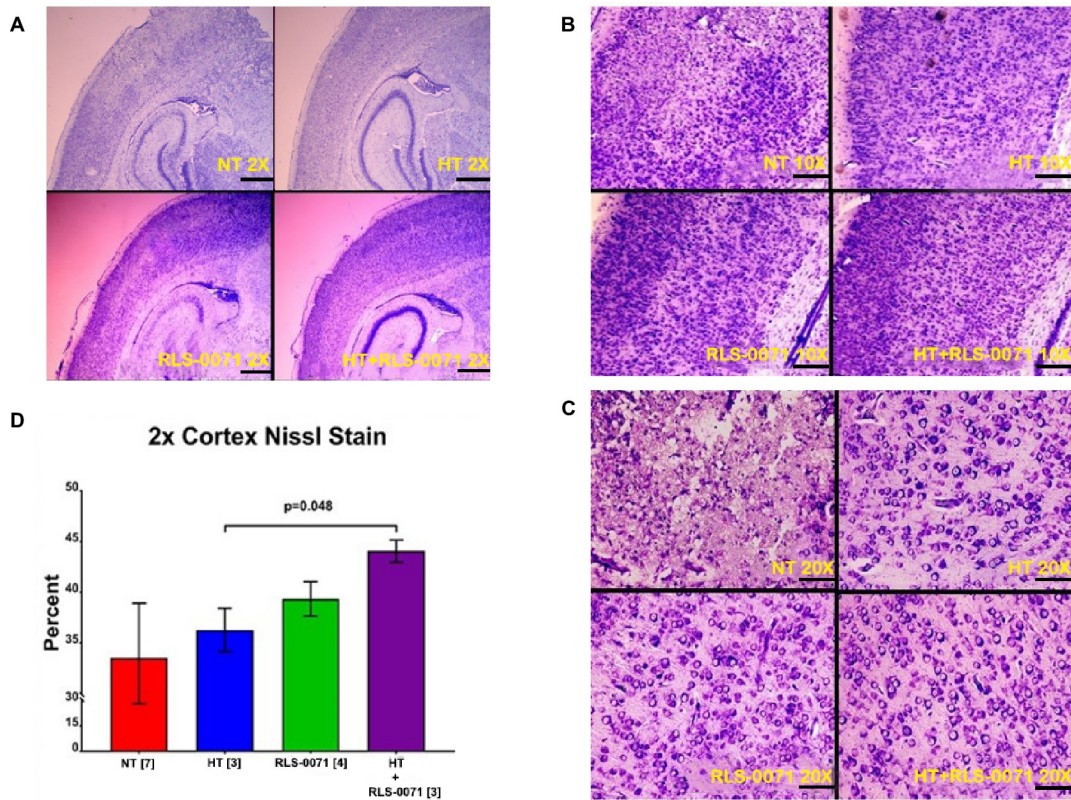

**Fig 2. Histological evaluation of cortical neuronal protection after HIE insult by RLS-0071 alone and in combination with hypothermia (HT).** Representative histological images of rat pup cortex Nissl bodies stained with Cresyl violet for pups euthanized at 48 hours after removal from hypoxia. In order of increasing magnification: 2 X (panel A), 10 X (panel B), 20X (panel C). Dense indigo tissue indicates a high density of neurons. The graph (panel D) shows percent cortical neuronal density measured by integrated density measurements performed for each group at 2× magnification. Data shown are means ± SEM. Groups: Normothermia (NT), hypothermia (HT), RLS-0071 given at 10 mg/kg ×2 doses (RLS-0071), hypothermia and RLS-0071 given at 10 mg/kg ×2 doses (HT+RLS-0071). Number of animals are shown in [].

HT + RLS-0071 also trended towards a lower amount of C1q compared with HT (P = 0.081). At the 8-hour timepoint, the HT group showed decreased C1q levels compared with NT (P = 0.002). Animals receiving HT +RLS-0071 also showed decreased C1q compared with NT (P = 0.005), however animals receiving RLS-0071 alone showed wide variability in brain C1q level resulting in no difference compared with NT. These data showed that RLS-0071 alone or RLS-0071 with hypothermia decreased C1q in the brain at the 1-hour timepoint, suggesting that RLS-0071 mediates changes in local C1q levels. These results suggest that RLS-0071 decreased C1q levels in the brain after hypoxia-ischemia insult, although there was some inconsistency.

## RLS-0071 and HT + RLS-0071 decreased brain injury as measured by neuroimaging

Magnetic resonance (MR) acquisitions were performed at 24 hours and 21 days after hypoxic/ischemic (H/I) insult to evaluate the effects of RLS-0071 on brain injury and tissue viability. A supplemental figure (S3 Fig) has been included showing representative coronal MRI images of the T2 map taken at 24 hrs after hypoxia comparing an animal from NT with another animal from HT+ RLS-0071 confirming the development of brain lesions in NT animals. Lesion T2

and Control T2 values (in milliseconds) were measured at 24h and 21d after H/I. T2-relaxation times give rise to the lesion contrast and represent increased tissue water caused by the local field dephasing effects that spins experience in the tissue. In intact, healthy tissues, the dephasing effects are larger (shorter T2-values) than in damaged tissues where the net-water is increased. S4 Fig shows the control side subjected to hypoxic injury alone with no changes seen in T2 value between study groups (pooled genders).

The 21 day follow up imaging allowed assessment of evolving brain injury and subsequent brain growth after injury in the setting of four interventions including NT, HT, RLS-0071 and HT + RLS-0071. Cerebral lesion volumes (mm³) measured at 24h and 21d after H/I show a significant reduction in lesion volume at 24 hours in animals receiving HT+ RLS-0071 compared to animals receiving hypothermic treatment alone (p<0.05) (Fig 3A). At 21 days the lesion volume again trends towards a decrease for HT + RLS-0071 compared with hypothermia alone, although not statistically significant. Lesion T2 (in milliseconds) were measured at 24h and 21d after H/I with T2-relaxation times are descriptive of tissue water in regard to local field dephasing effect spins experienced in the tissue. In intact, healthy tissues, the dephasing effects are larger (shorter T2-values) than in lesioned tissues where the net-water is increased. A significant reduction in T2 values is seen in animals receiving HT + RLS-0071 at day 21 compared to animals receiving hypothermia alone (p< 0.01) (Fig 3B). We also measured brain edema (%) at 24h and 21d after H/I. Edema was calculated by comparing the ipsilateral

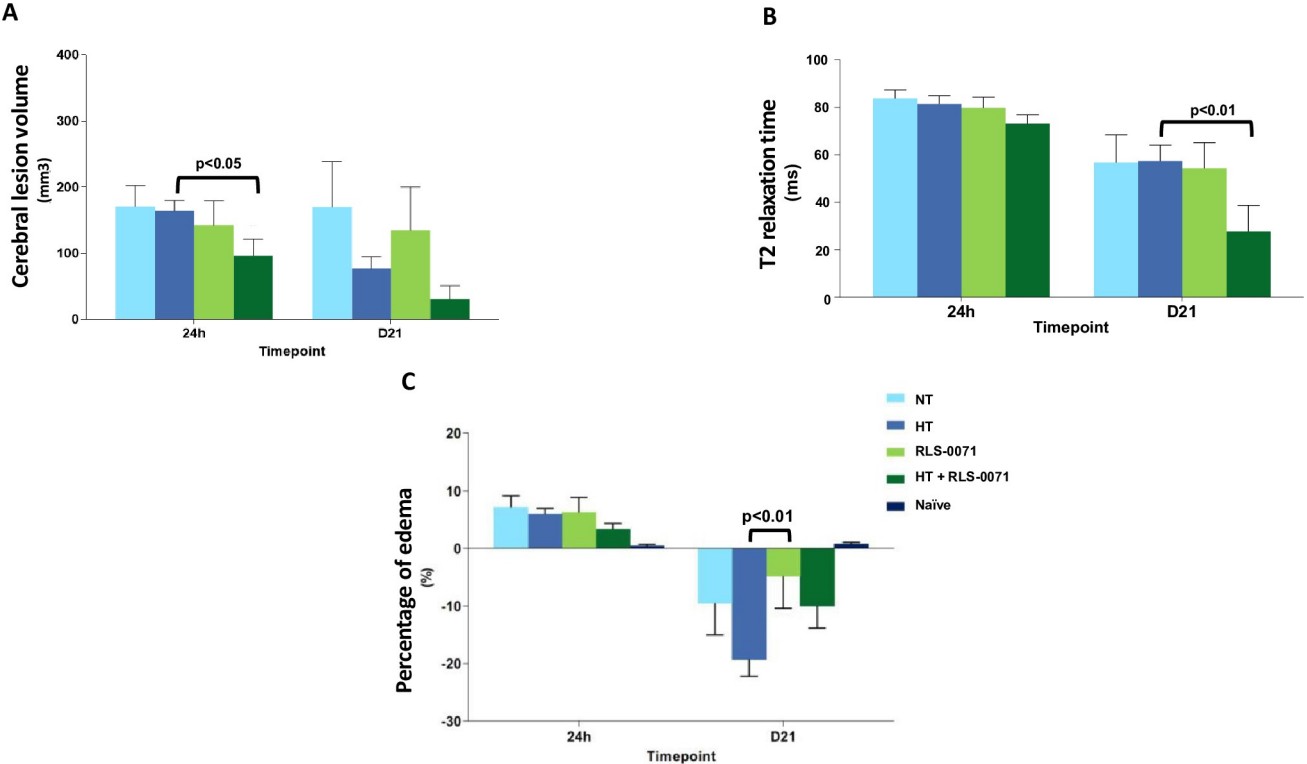

**Fig 3. T2-MRI for cerebral lesion volume and edema.** Panel A: The effects of normothermia/hypothermia and RLS-0071 treatment on cerebral lesion volume of neonatal Wistar rat pups subjected to brain hypoxia/ischemia (H/I). Data are presented as mean ± SEM. Panel B: The effects of normothermia/hypothermia and RLS-0071 treatment on lesion T2 values of neonatal Wistar rat pups subjected to brain hypoxia/ischemia (H/I). Data are presented as mean ± SEM. Panel C: The effects of normothermia/hypothermia and RLS-0071 treatment on edema percentage of neonatal Wistar rat pups (pooled genders) subjected to brain hypoxia/ischemia (H/I). Data are presented as mean ± SEM. Group 1: NT, n = 9; Group 2: HT, n = 10; Group 3: RLS-0071, n = 9; Group 4: HT+ RLS-0071, n = 9; Naïve, n = 5; n = number of animals.

hemisphere volume to the contralateral hemisphere. In the acute phase i.e 24 hrs, brain swelling results in positive edema values whereas values at later time points are measured as negative values due to restricted brain growth in the injured hemisphere compared with the contralateral hemisphere which continues to grow normally. Fig 3C shows that at 24h there is a trend towards decreased edema in the HT + RLS-0071 group of animals compared to the hypothermia alone (HT) animals, although not statistically significant. At day 21, animals receiving RLS-0071 under normothermia (RLS-0071) demonstrated a significant change in value indicating improved brain growth compared with animals receiving hypothermia alone (HT) (p <0.01). These studies show that RLS-0071 combined with hypothermia decreases cerebral lesion volume and T2 signal in animals subjected to H/I allowing for continued brain growth after H/I insult. These results suggest that pharmacological effect of RLS-0071 can decrease brain injury in this animal model of HIE.

## RLS-0071 and HT + RLS-0071 improved long term spatial memory retention in HIE

RLS-0071 preservation of neurons in the cortex at 48 hours after HIE procedures (Fig 2) suggested that neurocognitive outcomes may show improvement later in life. After allowing cohorts of HIE rats to grow to 6 weeks of age, we conducted a battery of neurocognitive tests. The Barnes maze tests long term spatial memory retention [29]. The Barnes maze latency time measures the total time the subjects took to find the escape hole whereas the Barnes maze errors test quantifies the number of errors, choosing false decoy holes, before burrowing in the escape hole were counted. The same 4 groups were tested: (1) NT, (2) HT, (3) RLS-0071, and (4) HT+RLS-0071. The HT group showed no improvement in latency time or errors compared with NT (Fig 4A). A trend towards decreased latency time was shown for the RLS-0071 (p = 0.083) and the HT+RLS-0071 (p = 0.019) groups compared with HT. Decreased errors

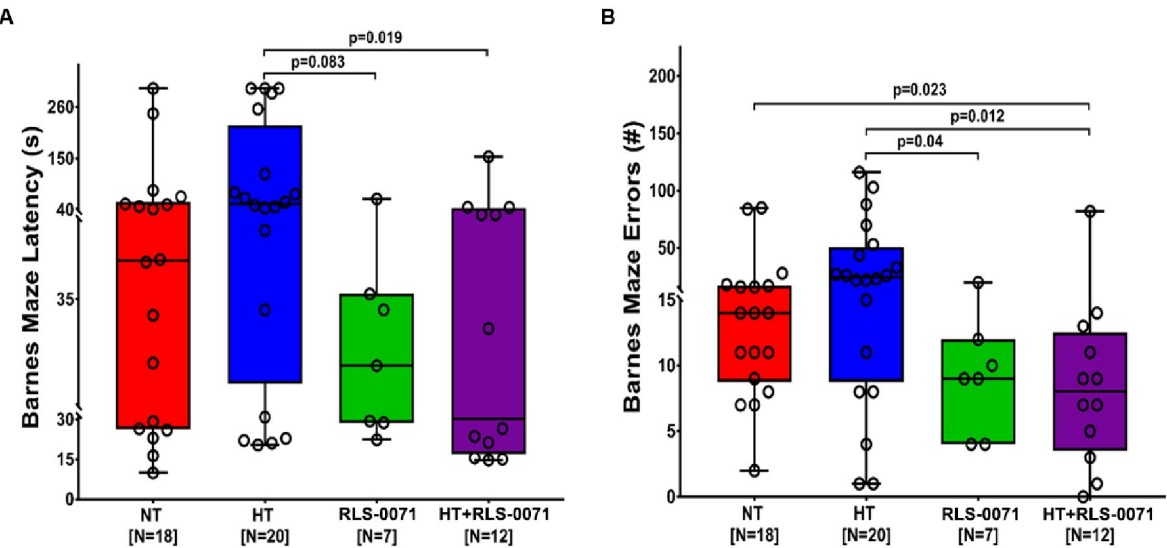

**Fig 4. Improvement in neurocognitive testing in young adult rats after HIE by RLS-0071 alone and in combination with hypothermia.** Panel A: Barnes maze latency measures the length of time required to find the escape hole. Data show each rat plotted as a circle with quartile boxes and 95th percentile whiskers. Panel B: Barnes maze errors measures the number of incorrect decoy holes investigated prior to finding the escape hole. Data show each rat plotted as a circle with quartile boxes and 95th percentile whiskers. Groups: Normothermia (NT), hypothermia (HT), RLS-0071 given at10 mg/kg ×2 doses (RLS-0071, hypothermia and RLS-0071 (HT+RLS-0071). Number of animals are shown in [].

was shown for the RLS-0071 (p = 0.04) and the HT+RLS-0071 (p = 0.012) groups compared with the HT group (Fig 4B). Together these data showed that RLS-0071 treatment groups had improved long term spatial memory compared with hypothermia alone.

### RLS-0071 and HT + RLS-0071 improve novel object recognition and locomotive function in HIE

Novel object recognition was performed to evaluate the influence of RLS-0071 on the animal's memory and cognition as a sensitive measure of altered animal behavior [30]. It serves as a test for long-term object memory performance and is a sensitive measurement for evaluating cognition [26]. The rats were given time to familiarize themselves with one object and then a new object was introduced. A rat that remembered the familiar object will spend more time investigating the novel object. The novel object recognition index is the ratio of time spent interacting with a familiar or novel object over the total combined familiar and novel object interaction time. Differences between the two indices demonstrates improved long-term memory. The same 4 groups were tested: (1) NT, (2) HT, (3) RLS-0071, and (4) HT+RLS-0071. There was no statistical difference between the medians for familiar and novel object indices for either NT or HT. Both RLS-0071 (p = 0.018) and HT+RLS-0071 (p = 0.01) showed significant differences between their indices, demonstrating improved long-term object memory (Fig 5A). These findings suggest that treatment with RLS-0071, with or without hypothermia, improved long term object memory compared with normothermia or hypothermia alone.

Rotarod task has been shown to be a sensitive and efficient measure of motor dysfunction after brain injury [27,31]. The rotarod device was used to assess motor coordination and balance using a small, suspended treadmill. Longer rotarod use times indicate better sensorimotor coordination. No significant difference in rotarod use times was seen between the four groups, suggesting no differences in sensorimotor coordination (Fig 5B).

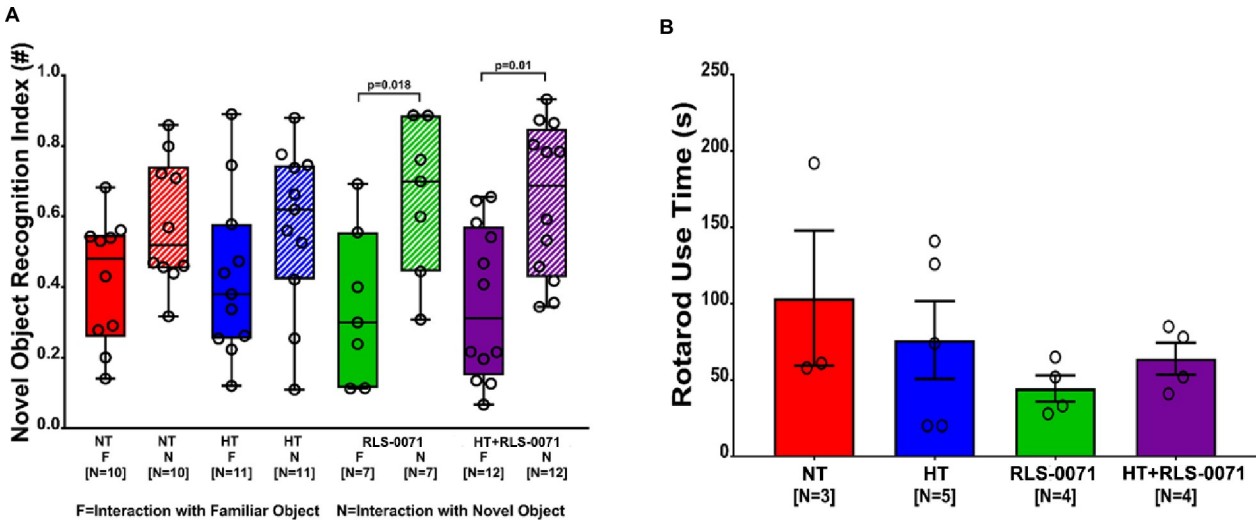

**Fig 5. Improvement in neurocognitive function in HIE by RLS-0071 alone and in combination with hypothermia.** Panel A: Novel object recognition index is a ratio of time spent with a familiar object (solid color) or novel object (hashed color) over total object interaction time. Increased time with the novel object demonstrates ability to remember the familiar object indicating normal long-term memory. Data show each rat plotted as a circle with quartile boxes and 95th percentile whiskers. Panel B: Rotarod testing evaluates sensorimotor coordination by measuring the length of time the subject can remain on the rod. Data show each rat plotted as a circle with quartile boxes and 95th percentile whiskers. Groups: Normothermia (NT), hypothermia (HT), RLS-0071 given at10 mg/kg ×2 doses (RLS-0071), hypothermia and RLS-0071 (HT+ RLS-0071). Number of animals are shown in [].

## Discussion

Therapeutic hypothermia (TH) is the current standard of care offered to neonates with HIE. It is the only intervention with proven benefit for decreasing mortality from HIE. Prior to the introduction of TH approximately 26.4% of infants who suffered HIE, but survived, experienced moderate to severe neurodevelopmental impairment and a further 14% survived with mild impairment. Reported rates of cerebral palsy following HIE are generally around 10%-13% among survivors of moderate to severe encephalopathy [32,33]. Although therapeutic hypothermia has improved the outlook for infants with moderate to severe HIE, with increased likelihood of survival without gross neurological abnormalities, it is important to note that learning deficits may be present with or without motor or sensory dysfunction [34]. Impairments in episodic memory associated with reduced hippocampal volume has been found in children following perinatal hypoxic-ischemic injury, but without associated neurological deficits [35]. Marlow et al [36] demonstrated memory and attention/executive function impairments in the severe encephalopathy group.

Currently no pharmacological interventions have been proven to benefit babies with HIE. The logistical difficulties of initiating therapeutic hypothermia within six hours after birth in order to yield a clinical benefit limits its use to babies born close to tertiary care neonatal intensive care units. Thus, many babies that are born geographically distant to such facilities are often deprived the benefit of therapeutic hypothermia. A pharmacological agent to treat HIE in any hospital around the world remains a major unmet medical need.

Previous studies have suggested that the classical complement pathway plays an important role in HIE pathogenesis [37,38]. During ischemia, neoantigens are expressed on endothelial cells and bind circulating natural IgM, leading to activation of the complement cascade via either the classical or lectin pathways and enhancing the pro-inflammatory response [39]. Recent studies suggest that therapeutic hypothermia may modulate complement activation and suggests an opportunity for complement modulating medicines to improve survival and neurocognitive outcomes for babies with HIE [14,15]. To date, no complement modulatory pharmacological interventions have been examined in HIE animal models. Our data with the dual target anti-inflammatory peptide, RLS-0071 with the ability to inhibit both humoral and cellular inflammatory processes demonstrated decreased brain damage in the standard rat HIE model. Starting at 24 hours after H/I insult, as shown on MRI imaging, RLS-0071 in conjunction with hypothermia reduced the cerebral lesion which correlated with histological analysis as demonstrated by increased percent of surviving neurons at 48 hours. RLS-0071 also improved early brain growth as demonstrated by neuroimaging at day 21. The neurocognitive findings in young adolescent rats subjected to hypoxia-ischemia as young pups suggest that RLS-0071 improved long-term neurocognitive outcomes as shown by better long-term memory.

We demonstrate that HT+ RLS-0071 offers improvement in memory (ability to remember familiar objects) and long-term spatial memory retention both of which are contributors towards enhanced learning. Together these studies show that a complement inhibitor peptide targeting C1 with antioxidant and neutrophil modulating effects can provide additional neuroprotection over hypothermia alone when given as rescue therapy in an animal model of HIE.

From a mechanism of action standpoint, these studies suggest that a pharmacological intervention that inhibits classical complement pathway activation with antioxidant and myeloperoxidase and NETosis inhibitory effects could hold promise for modifying HIE brain damage.

Given the recent change in our understanding of the additional functions of RLS-0071, future studies will explore the antioxidant and neutrophil modulating effects of RLS-0071 in the HIE animal model further elucidating the roles of these newly described aspects of HIE

pathogenesis. These studies are beyond the scope of this current manuscript where we focus on the impact of RLS-0071 on HIE related brain damage and neurocognitive function. Future studies will include administering RLS-0071 at varying time points after hypoxic-ischemic injury to determine if rescue from HIE can be extended out beyond the 1 hour we have currently shown. We plan to also include follow up neuro imaging and histological studies of the developing brain through childhood and adolescence in the animal model.

## Supporting information

**S1 Fig. Lowest effective treatment dose of RLS-0071 for HIE.** Rat pups were treated with different doses of RLS-0071 (also known as PIC1) at one hour post hypoxia with a repeat dose 4 hours later or untreated (normothermia. NT). RLS-0071 doses: 10 mg/kg × 2, 5 mg/kg × 2, 1 mg/kg × 2 were piloted. Animals were euthanized 48 hours after hypoxia, brains extracted, photographed and evaluated for any evidence of gross brain infarction. Panel A: The image on the left shows a normal appearing brain after treatment with PIC1 (10 mg/kg × 2). The image on the right shows the brain from an animal that did not receive RLS-0071 kept at normothermia (NT). The area of infarction is circled in yellow. Panel B: The graph shows the percent of animals showing any gross evidence of brain infarction for each group.
(PDF)

**S2 Fig. Brain measurements of C1q after HIE.** C1q measured by ELISA on brain homogenates were prepared after euthanasia at 1 hour or 8 hours after the second RLS-0071 dosing. Data shown are means ± SEM. Groups: normothermia (NT), hypothermia (HT), RLS-0071 10 mg/kg ×2 doses (RLS-0071), hypothermia and RLS-0071 (HT+RLS-0071). Number of animals are shown in [].
(PDF)

**S3 Fig. MRI acquisitions performed at 24h after hypoxia/ischemia injury.** Representative figure of the effects of normothermia and hypothermia +RLS-0071 treatment on lesion T2 values of neonatal Wistar rat pup subjected to brain hypoxia/ischemia (H/I) measured at 24 hours confirm the developed lesion in NT animals. Groups: normothermia (NT) and RLS-0071 with hypothermia (HT+RLS-0071).
(PDF)

**S4 Fig. T2 MRI relaxation time.** The effects of normothermia, hypothermia and RLS-0071 treatment (with and without hypothermia) on control T2 values of neonatal Wistar rat pups (pooled gender) subjected to brain hypoxia/ischemia (H/I). Data are presented as mean ± SEM. Group 1: NT, n = 9; Group 2: HT, n = 10; Group 3: RLS-0071, n = 9; Group 4: HT+ RLS-0071, n = 9; Naïve, n = 5; n = number of animals.
(PDF)

## Author Contributions

**Conceptualization:** Kenji Cunnion, Neel Krishna.

**Data curation:** Parvathi Kumar, Kenji Cunnion.

**Investigation:** Pamela Hair, Kenji Cunnion.

**Methodology:** Pamela Hair.

**Project administration:** Kenji Cunnion.

**Resources:** Neel Krishna.

**Software:** Parvathi Kumar.

**Supervision:** Kenji Cunnion, Neel Krishna, Thomas Bass.

**Visualization:** Kenji Cunnion.

**Writing – original draft:** Parvathi Kumar.

**Writing – review & editing:** Parvathi Kumar, Pamela Hair, Kenji Cunnion, Neel Krishna, Thomas Bass.

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
