## [Decision Letter · Decision Letter 0]

10 Jun 2021

PONE-D-21-14373

Classical Complement Pathway Inhibition Reduces Brain Damage in a Hypoxic Ischemic Encephalopathy Animal Model

PLOS ONE

Dear Dr. Kumar,

Thank you for submitting your manuscript to PLOS ONE. After careful consideration, we feel that it has merit but does not fully meet PLOS ONE’s publication criteria as it currently stands. Therefore, we invite you to submit a revised version of the manuscript that addresses the points raised during the review process.

Methodological aspects and discussion are the two sections that need to be mostly improved.

We look forward to receiving your revised manuscript.

Kind regards,

Olivier Baud, MD, PhD

Academic Editor

PLOS ONE

 [PK, PH, NKK, KC are employees of ReAlta life sciences. The funders had no role in study design, data collection and analysis, decision to publish, or preparation of the manuscript.].

We note that one or more of the authors is affiliated with the funding organization, indicating the funder may have had some role in the design, data collection, analysis or preparation of your manuscript for publication; in other words, the funder played an indirect role through the participation of the co-authors. If the funding organization did not play a role in the study design, data collection and analysis, decision to publish, or preparation of the manuscript and only provided financial support in the form of authors' salaries and/or research materials, please do the following:

a. Review your statements relating to the author contributions, and ensure you have specifically and accurately indicated the role(s) that these authors had in your study. These amendments should be made in the online form.

b. Confirm in your cover letter that you agree with the following statement, and we will change the online submission form on your behalf:

“The funder provided support in the form of salaries for authors [insert relevant initials], but did not have any additional role in the study design, data collection and analysis, decision to publish, or preparation of the manuscript. The specific roles of these authors are articulated in the ‘author contributions’ section.

3. Thank you for providing the following Funding Statement: 

[I have read the journal's policy and the authors of this manuscript have the following competing interests: PK, PH, NKK, KC are employees of ReAlta Life Sciences Inc.].

We note that one or more of the authors is affiliated with the funding organization, indicating the funder may have had some role in the design, data collection, analysis or preparation of your manuscript for publication; in other words, the funder played an indirect role through the participation of the co-authors.

If the funding organization did not play a role in the study design, data collection and analysis, decision to publish, or preparation of the manuscript and only provided financial support in the form of authors' salaries and/or research materials, please review your statements relating to the author contributions, and ensure you have specifically and accurately indicated the role(s) that these authors had in your study in the Author Contributions section of the online submission form. Please make any necessary amendments directly within this section of the online submission form.  Please also update your Funding Statement to include the following statement: “The funder provided support in the form of salaries for authors [insert relevant initials], but did not have any additional role in the study design, data collection and analysis, decision to publish, or preparation of the manuscript. The specific roles of these authors are articulated in the ‘author contributions’ section.”

If the funding organization did have an additional role, please state and explain that role within your Funding Statement.

Please also provide an updated Competing Interests Statement declaring this commercial affiliation along with any other relevant declarations relating to employment, consultancy, patents, products in development, or marketed products, etc. 

4. We note that Figure 2 (A, C, D) and S1 Figure A in your submission contain copyrighted images. All PLOS content is published under the Creative Commons Attribution License (CC BY 4.0), which means that the manuscript, images, and Supporting Information files will be freely available online, and any third party is permitted to access, download, copy, distribute, and use these materials in any way, even commercially, with proper attribution. For more information, see our copyright guidelines: http://journals.plos.org/plosone/s/licenses-and-copyright.

a. You may seek permission from the original copyright holder of Figure 2 (A, C, D) and S1 Figure A to publish the content specifically under the CC BY 4.0 license.

5. Please ensure that you refer to Figure 6 in your text as, if accepted, production will need this reference to link the reader to the figure.

Additional Editor Comments (if provided):

Reviewers' comments:

Reviewer's Responses to Questions

**Comments to the Author**

1. Is the manuscript technically sound, and do the data support the conclusions?

Reviewer #1: Partly

Reviewer #2: Yes

Reviewer #3: Partly

2. Has the statistical analysis been performed appropriately and rigorously? 

Reviewer #1: No

Reviewer #2: Yes

Reviewer #3: No

3. Have the authors made all data underlying the findings in their manuscript fully available?

Reviewer #1: Yes

Reviewer #2: No

Reviewer #3: No

4. Is the manuscript presented in an intelligible fashion and written in standard English?

Reviewer #1: Yes

Reviewer #2: Yes

Reviewer #3: Yes

5. Review Comments to the Author

Reviewer #1: The authors investigate the effect on an association between the classical complement inhibitor RLS-0071 and hypothermia treatment in a rat model of HIE. The results are interesting and evidence an effect of RLS-0071 alone or in association with hypothermia.

I have however some concern and a major revision of the manuscript should be done. Below my comments in detail

1 Please clarify and explain which time points were investigated. In the abstract section the authors talked about 1day, 21 days, 1h and 8h whereas in the results section results related to 16h and 48h are reported.

2 Did the authors investigated only male or both sex

3 The groups annotation in the results section is quite confusion they talk about group 1,2,3... and annotation with the real treatment will make the results more understandable

4 The dosage and the administration route of RLS-0071 was modified in the neuroimaging studies. Why? Please explain and justify. Which vehicle was used? PBS, DMSO? RLS-0071 injection and beginning of hypothermia were sequential?

5 Line 144 right hemispheres of brains..... why only right?

Major comment

The experimental n is not always sufficient to support the observation reported. HIE model is a variable model due to the variation of lesion size and severity of brain injury. Some of the results are supported by experiment performed with N=3, 4 or 5 whereas for other the authors reported n between 7 and 12. Further analysis should be done to increase the experimental number

Reviewer #2: The paper by Kumar and colleagues aimed to investigate the role of a complement pathway inhibitor on short- and long-term outcome in newborn rats. The authors used an established neonatal rat model of unilateral hypoxic-ischemic brain injury to answer their research question. In addition authors used a battery of different read-out parameters (histology, protein analysis, MRI, neurobehavioral outcome) to answer their research question.

The paper and research question is of potential interest, however the paper is lacking methodological power and the numbers of animals may be too low to answer some of the raised questions. My major concern is the lack of hypothermic neuroprotection in the model presented by Kumar and the authors should explain and discuss this. Additionally, the potential neuroprotection by RLS-0071 may not only due to complement mode of action, as the altered levels of complement are not satisfying.

In general, the paper needs major revision and I hope to improve the quality of the paper with my comments and suggestions.

Methods:

- When were the experiments performed, P10, 11 or 12? How many animals were used at which postnatal age? Was there a difference in outcome? Brain maturation and susceptibility to HI will be different between P10 and 12.

- Where does the RLS-0071 dosage come from?

- Was target temperature measured in all pups? What was the target temperature in the NT group? What was the temperature during the insult? 45min seems a quite short time in P10 rats.

- It says the rats underwent HT for 6 hours, however if given RLS-0071 one and four hours following hypoxia and then given back to the dam, this is only 5 hours. Please explain.

- In Figure 1 it says recovery for 60 mins after hypoxia. So did the treatment start with 60min delay? If yes, why and where were the pups kept between and at which temperature?

- How many animals were used per group, please include into figure 1

- Why was no HI+saline ip group included?

- Line 114 says Pentobarbital was used, line 134-140 does not include pentobarbital. Please explain.

Statistics:

- Why was mean and SEM used and not median and IQR? Was the data normally distributed? Can the authors show dot blots with the result of each individual animal, so the reader can see the distribution of inury?

Results:

- For S1A/B only n=3 or 4 animals were used. This seems very low number with the known large variability in this model. Can the authors please show dot blots? Do you have coronal MRI slices showing the infarction?

- Figure 1: did the authors also analyse different brain areas, e.g. hippocampus? If not, why not? Where in the cortex was the analysis performed, how many fields of view? Again, number of animals appears low, please explain.

- Why was HT not neuroprotective in this model? Maybe too little number of animals?

- Figure 3: is is the large variability due to the small sample sizes and model variability?

- How many animals had MRI perfomed? Was it the same animal at day 1 and 21?

- Why was hypothermia not neuroprotective measured by long term outcome? This is very surprising given the known literature.

Discussion:

- Please comment on the role of RLS-0071 as free radical scaveneger and the possibility of mechanism of action for the shown results in the study

- The discussion is too short and superficial and should be improved.

Minor:

- Line 28: HIE instead of HIA

-

Reviewer #3: Reviewer

In the study PONE-D-21-14373, Kumar and colleagues tested the effects a classical complement pathway inhibitor (RLS-0071) associated, or not, with hypothermia in the Vannucci model of neonatal HI in Wistar rats. Authors state that adjunctive therapy targeting inhibiting complement system improves the benefits afforded by therapeutical hypothermia. In general, the study has a clear hypothesis, the references used to sustain the results are adequate, however the study needs improvements in the description of the methods, results and a real discussion of the findings (which is not present in this version) and clear conclusion message. Despite the good level of grammar, some sentences need to be reorganized in order to make more understandable for the readers.

General Comments

# The legends of the Figures should be improved and moved to the end of the manuscript, since it is difficult to follow the sequence of the manuscript.

# The general scheme of administration, and especially the RLS-0071 in the HT groups should be better explained in the methods section.

# The Discussion section need to be re-written and discuss in deep the data obtained in the study.

Abstract and Intro

# Line 28 - The short name for Hypoxia-ischemia should be HIE.

# HI animals received 4 interventions? “one of the 4” is more adequate.

# Please consider using the same terminology (RLS-0071) or PIC1 in the whole manuscript;

Methods

# In the session materials (l 91 to 94), please describe the code of antibodies used.

# The authors perform the surgery from P10 to P12. However, due to the nature of the lesion, as well as brain energetic metabolism differences that can emerge from this range, do the authors have a control for the variability of the lesion? Despite the observed variability observed in the biochemical analysis, it is possible to observe it in the Rota-Rod test (Figure 6B - NT group).

# It is important to mention in the section Animal model, the hemisphere used in the study and the sex of the animals.

# Were the sentinel animals included in the study? Some authors highly recommend to remove these animals from the analysis due to the stress caused by the rectal monitoring for this prolonged period.

# How was the rewarming procedure? Did it take the 60 minutes recovery? Was the return to normothermia constant? Please, describe in more details.

# The description of sham group (l. 130) is not correct according what is a sham group (all procedures, without injury). It should be called as naïve, if there is no manipulation (line 131 and 132).

# Do the authors consider using only “sham” animals as the controls for the lesion in the MRI study, since a large number of studies” state that the contralateral hemisphere is not exactly a control? Is it possible to use Sham+normo as the control group?

# For the neuroimaging studies animals were injected subcutaneously, what is the reason?

# Please add the references for used in the Neurobehavioral tests.

# The difference in the number of animals used for the behavioral analysis has an explanation? Was there a cut-off for the Rota-Rod test?

# Regarding the Novel Object Recognition test, authors performed the test in the same day of second training session, or next day? If it is 24h interval period, authors should consider long-term instead short-term memory evaluation.

Results

# Figure 1 - Please add a scale bar.

# How homogeneous was the injury procedure? Was the number of animals (Fig S1) enough to reach this initial assumption of tissue protection? No statistical analysis was performed? Could you use an accepted classification of injury? The one proposed by by Palmer 1992 (003 1-3998/93/3304-0405S03.00/0)?

# Did author have an explanation for such variability in C1q levels in the brain at 8h? it seems from the hemolysis evaluation (Figure S1 ) that the variability of the inhibitor is very small until 8h. The variability of the model explains this??

# Figure 3a - Please refer, to Brain C1q levels, instead to Cranial.

# The neuroimaging results, in which brain “growth” was stated, could be re-written. Normal brain volume, perhaps?

# Line 413…. Despite no statistical difference (because it is 0.08), a trend to decreased latency….

# Is there a difference among the 4 groups in NOR regarding the exploration of familiar object?

# the description of Rota rod as an ability test, is very imprecise. RR tests animals’ coordination, balance, learning but not agility…Please, review. Despite this, the results (3 animals in NT and 5 in HI) clearly show the variability of the model, which could bias the final conclusions of the study.

Despite the very promising results from the group, based in the data here presented, I could not recommend the manuscript for publication in the present format.

6. PLOS authors have the option to publish the peer review history of their article (what does this mean?). If published, this will include your full peer review and any attached files.

Reviewer #1: No

Reviewer #2: No

Reviewer #3: **Yes: **Eduardo Farias Sanches

---

## [Author Response · Author response to Decision Letter 0]

22 Jul 2021

Journal Requirements:

Authors response: We have revised the manuscript to fit the journal requirements.

 [PK, PH, NKK, KC are employees of ReAlta life sciences. The funders had no role in study design, data collection and analysis, decision to publish, or preparation of the manuscript.].

We note that one or more of the authors is affiliated with the funding organization, indicating the funder may have had some role in the design, data collection, analysis or preparation of your manuscript for publication; in other words, the funder played an indirect role through the participation of the co-authors. If the funding organization did not play a role in the study design, data collection and analysis, decision to publish, or preparation of the manuscript and only provided financial support in the form of authors' salaries and/or research materials, please do the following:

a. Review your statements relating to the author contributions, and ensure you have specifically and accurately indicated the role(s) that these authors had in your study. These amendments should be made in the online form.

b. Confirm in your cover letter that you agree with the following statement, and we will change the online submission form on your behalf:

“The funder provided support in the form of salaries for authors [insert relevant initials], but did not have any additional role in the study design, data collection and analysis, decision to publish, or preparation of the manuscript. The specific roles of these authors are articulated in the ‘author contributions’ section.

Authors response:

The funder provided support in the form of salaries for authors [PK,PH,KC, NK], but did not have any additional role in the study design, data collection and analysis, decision to publish, or preparation of the manuscript. The specific roles of these authors are articulated in the ‘author contributions’ section.

3. Thank you for providing the following Funding Statement: 

[I have read the journal's policy and the authors of this manuscript have the following competing interests: PK, PH, NKK, KC are employees of ReAlta Life Sciences Inc.].

We note that one or more of the authors is affiliated with the funding organization, indicating the funder may have had some role in the design, data collection, analysis or preparation of your manuscript for publication; in other words, the funder played an indirect role through the participation of the co-authors.

If the funding organization did not play a role in the study design, data collection and analysis, decision to publish, or preparation of the manuscript and only provided financial support in the form of authors' salaries and/or research materials, please review your statements relating to the author contributions, and ensure you have specifically and accurately indicated the role(s) that these authors had in your study in the Author Contributions section of the online submission form. Please make any necessary amendments directly within this section of the online submission form. Please also update your Funding Statement to include the following statement: “The funder provided support in the form of salaries for authors [insert relevant initials], but did not have any additional role in the study design, data collection and analysis, decision to publish, or preparation of the manuscript. The specific roles of these authors are articulated in the ‘author contributions’ section.”

If the funding organization did have an additional role, please state and explain that role within your Funding Statement.

Please also provide an updated Competing Interests Statement declaring this commercial affiliation along with any other relevant declarations relating to employment, consultancy, patents, products in development, or marketed products, etc. 

Authors response: Please find included both an updated Funding Statement and Competing Interests Statement. Thank you for changing the online submission form on our behalf.

The funder provided support in the form of salaries for authors [PK,PH,KC, NK], but did not have any additional role in the study design, data collection and analysis, decision to publish, or preparation of the manuscript. The specific roles of these authors are articulated in the ‘author contributions’ section.”

Competing Interests Statement:

Commercial affiliation with ReAlta Life Sciences Inc, does not alter our adherence to PLOS ONE policies on sharing data and materials.

4. We note that Figure 2 (A, C, D) and S1 Figure A in your submission contain copyrighted images. All PLOS content is published under the Creative Commons Attribution License (CC BY 4.0), which means that the manuscript, images, and Supporting Information files will be freely available online, and any third party is permitted to access, download, copy, distribute, and use these materials in any way, even commercially, with proper attribution. For more information, see our copyright guidelines: http://journals.plos.org/plosone/s/licenses-and-copyright.

a. You may seek permission from the original copyright holder of Figure 2 (A, C, D) and S1 Figure A to publish the content specifically under the CC BY 4.0 license.

Authors Response: 

Regarding S1 Figure A, We are not aware where this figure was published or used at. Furthermore, S1 Fig A and B must be shown simultaneously for the data to be interpretable and useful. After correspondence (Case number: 07205094) with the editorial assistant at PLOS One we have elected to not make any changes to the figure. 

After careful and extensive review of publication by Shah et al (1) Figure 2 with potential copyright images have been replaced with new images include with current submission.

5. Please ensure that you refer to Figure 6 in your text as, if accepted, production will need this reference to link the reader to the figure.

Authors Response: Thank you for pointing out the oversight. Figure 6 has been referred to in the corrected manuscript.

Additional Editor Comments (if provided):

Reviewers' comments:

Reviewer's Responses to Questions

Comments to the Author

1. Is the manuscript technically sound, and do the data support the conclusions?

Reviewer #1: Partly

Reviewer #2: Yes

Reviewer #3: Partly

2. Has the statistical analysis been performed appropriately and rigorously?

Reviewer #1: No

Reviewer #2: Yes

Reviewer #3: No

3. Have the authors made all data underlying the findings in their manuscript fully available?

Reviewer #1: Yes

Reviewer #2: No

Reviewer #3: No

4. Is the manuscript presented in an intelligible fashion and written in standard English?

Reviewer #1: Yes

Reviewer #2: Yes

Reviewer #3: Yes

5. Review Comments to the Author

Reviewer #1: The authors investigate the effect on an association between the classical complement inhibitor RLS-0071 and hypothermia treatment in a rat model of HIE. The results are interesting and evidence an effect of RLS-0071 alone or in association with hypothermia.

I have however some concern and a major revision of the manuscript should be done. Below my comments in detail

1 Please clarify and explain which time points were investigated. In the abstract section the authors talked about 1day, 21 days, 1h and 8h whereas in the results section results related to 16h and 48h are reported.

Authors Response: Thank you for pointing out the discrepancy. We have streamlined the abstracts with the results. The statement below has been added to the manuscript as well. 

Brain examination for gross infarction and histology was performed after euthanasia at 48 hours post-hypoxia. Prior to euthanasia, cohorts for neurocognitive testing were allowed to grow to young adult age and then were tested for long-term spatial memory retention, novel object recognition, and locomotive function.

MRI acquisitions were performed at 24h after surgical procedure to confirm the developed lesion. Additional, T2-volumetric MRI was performed 21 days after surgery to measure for infarct volume and edema in all rats.

2 Did the authors investigated only male or both sex

Authors Response:

Both sexes were included in the investigation. This has been clarified in the animal model description.

3 The groups annotation in the results section is quite confusion they talk about group 1,2,3... and annotation with the real treatment will make the results more understandable

Authors Response:

Thank you for the feedback. We have referred to the groups to animals with the treatment administered. 

4 The dosage and the administration route of RLS-0071 was modified in the neuroimaging studies. Why? Please explain and justify. Which vehicle was used? PBS, DMSO? RLS-0071 injection and beginning of hypothermia were sequential?

Authors Response:

The neuroimaging studies were done by a commercial vendor, Charles River Discovery Research Services Finland Ltd. RLS-0071 was provided as ‘ready to use’ formulations to the vendor and the drug was administered as a fixed volume of administration at 200�l subcutaneous per institutional policy. 

The dosing at 12.5mg/kg x2 doses was roughly similar to that used for the studies done by the primary lab at ReAlta Life sciences at 10mg/kg x 2 doses. The difference in dosage was noticed as an after sight after the completion of the neuroimaging study. The route of administration was switched from intraperitoneal to subcutaneous administration due to the veterinarian policy at the commercial vendor site.

The vehicle used was 0.05 M Histidine buffer pH 6.7. This information has been added to the manuscript as well.

The first dose of RLS-0071 was given immediately after the recovery phase post hypoxia and just prior to start of hypothermia to time it along with the rescue therapy of therapeutic hypothermia.

5 Line 144 right hemispheres of brains..... why only right?

Authors Response:

The animal model causes unilateral ischemic injury to the animal brain. In the case of the experiments described in the manuscript, the injury was always induced by ligation of the right carotid artery which resulted in a right sided ischemic brain injury thus only the right side of the brain was used for histopathology and brain C1q measurements. 

Major comment

The experimental n is not always sufficient to support the observation reported. HIE model is a variable model due to the variation of lesion size and severity of brain injury. Some of the results are supported by experiment performed with N=3, 4 or 5 whereas for other the authors reported n between 7 and 12. Further analysis should be done to increase the experimental number

Authors response: We appreciate the reviewer’s concern about our statistical approach and the variability inherent to the animal model described. 

To limit variability, all of the hypoxia/ischemia injury inducing surgeries at the primary lab in ReAlta (i.e results generating the results pertaining to gross examination, histology and cranial C1q ) were performed in Wistar rat pups by the same individual who was trained in the procedure by Susan Vannucci and through the course of these experiments performed nearly 200 procedures in the same lab set up with the same instruments in under 8 minutes with high reliability and on target performance. This allowed for us to calibrate the procedure to reliably cause injury where an impact could be measured by the interventions RLS-0071 and HT. 

In the analysis of human data an n = 4 would be inadequate due to the high degree of variability of genetic background, age, diet, lifestyle, etc. However, in experiments utilizing rats of a single strain, housed under the same conditions, eating the same diet and undergoing identical procedures, the variability is highly constrained, and this is reflected in the tight error bars seen for these data. Non-parametric tests are warranted for small sample sizes where there is reason to doubt that the data conforms with a normal distribution. Our data show limited evidence of skew and we have no reason to suspect that the data is not normally distributed. There is also the ethical consideration against conducting animal experiments with large numbers of animals when much smaller numbers (e.g. n = 4) can yield meaningful results. Please see the following reference detailing the argument for utilizing the minimal numbers of animals possible and t-tests in animal experimentation (Michael F. W. Festing, Douglas G. Altman, Guidelines for the Design and Statistical Analysis of Experiments Using Laboratory Animals, ILAR Journal, Volume 43, Issue 4, 2002, Pages 244–258, https://doi.org/10.1093/ilar.43.4.244).

Reviewer #2: The paper by Kumar and colleagues aimed to investigate the role of a complement pathway inhibitor on short- and long-term outcome in newborn rats. The authors used an established neonatal rat model of unilateral hypoxic-ischemic brain injury to answer their research question. In addition authors used a battery of different read-out parameters (histology, protein analysis, MRI, neurobehavioral outcome) to answer their research question.

The paper and research question is of potential interest, however the paper is lacking methodological power and the numbers of animals may be too low to answer some of the raised questions. My major concern is the lack of hypothermic neuroprotection in the model presented by Kumar and the authors should explain and discuss this. Additionally, the potential neuroprotection by RLS-0071 may not only due to complement mode of action, as the altered levels of complement are not satisfying.

In general, the paper needs major revision and I hope to improve the quality of the paper with my comments and suggestions.

Methods:

- When were the experiments performed, P10, 11 or 12? How many animals were used at which postnatal age? Was there a difference in outcome? Brain maturation and susceptibility to HI will be different between P10 and 12.

Authors Response:

P10 was used since it is considered equivalent to human term newborn (2). At P10, hypoxic ischemic brain injury was induced using the Vannucci method of unilateral carotid ligation (3) in keeping with published literature (4).

- Where does the RLS-0071 dosage come from?

Authors Response:

Early studies with RLS-0071 in rats suggested that doses of 160 mg/kg were needed to produce complement inhibition in the bloodstream. Based on the working drug dosing rationale at the time ‘that complement inhibition in the bloodstream of the rat would be necessary for brain protection’ our earliest studies in a rat model of hypoxic ischemic encephalopathy (HIE) were conducted with RLS-0071 at doses of 200 – 400 mg/kg given IP as two divided doses over 4 hours. These doses demonstrated decreased zones of brain infarction compared with no treatment or hypothermia alone, the standard of care. We tested doses of RLS-0071 as high as 800 mg/kg without seeing additional benefit in decreasing brain infarction. Subsequent experiments were performed to identify the minimal effective dose and we eventually determined that much lower doses of RLS-0071 (i.e. 10 mg/kg IP × 2 doses) would yield equivalent brain protection to doses of 160 mg/kg.

- Was target temperature measured in all pups? What was the target temperature in the NT group? What was the temperature during the insult? 45min seems a quite short time in P10 rats.

Authors Response:

The rectal temperature of sentinel pups were monitored to ensure normothermic body temperature during hypoxia exposure.

Normothermia was maintained at 37°C +/- 1 rat internal temperature.

Various labs have performed varying modifications of the hypoxia inducing procedures in these animals models (5, 6). As shown below (figure 1,2,3 ), we elected to test varying times of hypoxia insult. Due to the non-statistical difference in effects measured, it was decided to subject the animals to the least amount of hypoxia (45 minutes) which can reliably cause injury objectively measured by MRI.

The brain tissue viability of H/I operated Wistar rat pups, at the age of 10 days, subjected to total hypoxia are presented in Figure 1. Lesion T2 and Control T2 values were measured at 24h after H/I, and the data are shown as average. Please note, the higher the relaxation time, the more water content (due to cell swelling, cytotoxic/vasogenic oedema, CSF leakage or any other process), the LESS viable the tissue. 

According to the conducted one-way ANOVA, 45/60/75 minutes of total hypoxia did not show any significant difference in the T2 relaxation times of these animals (p > 0.05, one-way ANOVA).

Figure 2 : According to the conducted one-way ANOVA, 45/60/75 minutes of total hypoxia did not show any significant difference in the lesion volumes of these animals (p > 0.05, one-way ANOVA)

Figure 3: According to the conducted one-way ANOVA, 45/60/75 minutes of total hypoxia did not show any significant difference in the edema percentages of these animals (p > 0.05, one-way ANOVA)

- It says the rats underwent HT for 6 hours, however if given RLS-0071 one and four hours following hypoxia and then given back to the dam, this is only 5 hours. Please explain.

Authors Response:

RLS-0071 was given to the animals starting at 1 hr after hypoxia injury and repeat dose given again at 4 hours during the hypothermia treatment phase. At the end of 6 hours of hypothermia, the animal was rewarmed and returned to the dam. This has been clarified in Figure 1.

The wording below has been incorporated into the manuscript:

‘When RLS-0071 was administered, it was administered as 2 doses of 10mg/kg, intraperitoneal (i.p.) given 4 hours apart starting 60 minutes after removal from the hypoxia chamber. The second dose of RLS-0071 was administered while the animals were in the hypoxia chamber, 4 hours after the first dose. After each intervention, the pups were rewarmed to 37±1oC on a homeothermic blanket system.’

- In Figure 1 it says recovery for 60 mins after hypoxia. So did the treatment start with 60min delay? If yes, why and where were the pups kept between and at which temperature?

Authors Response:

The treatment with RLS-0071 and/or hypothermia was started after 60 minutes recovery from hypoxic insult per IACUC suggestion for post surgery recovery and survival. This allowed for establishment of the described animal model (4). Of note, as noted by Shah et al, systemic C3a expression in the hypothermia group was increased starting at 1 hr after the hypoxic insult which was the time we targeted for optimal neuroprotective effect as an additive to the current standard of therapeutic hypothermia. Hypothermia has been established to inhibit the expression of C1q (i.e., clearance of apoptotic cells and classical pathway activation), C3-fragments (i.e., opsonins) and C9 (i.e., membrane attack complex) in brain tissue after hypoxia, suggesting that inhibition of these complement effectors plays a role in hypothermia neuroprotection. RLS-0071 furthers this interference with the cytotoxic cascade through its known activity including classical complement pathway inhibition and anti-oxidant effects.

During the 60-minute time interval between end of hypoxia and start of therapy with RLS-0071 and hypothermia, the animal was allowed to recover from anesthesia and rewarmed to 37±1oC. Following hypothermia treatment, the animal was rewarmed to 37±1oC on a homeothermic blanket system. Each animal used varying times to recover but all of them recovered in 60 minutes. When all the animals had recovered, they were returned to the nursing dam en masse and euthanized at prespecified time points.

- How many animals were used per group, please include into figure 1

Authors Response:

Thank you for the suggestion but we politely disagree with the recommendation. Figure 1 is intended to demonstrate the general sequence of events performed in the animal model and the time course of the various interventions administered.

All the experiments performed to generate these figures were done spread over 12-18 months in 2 separate set ups. The neuroimaging and neurocognitive testing were done in Finland by a commercial vendor while the initial pilot, dose finding studies were done by the primary lab at ReAlta. Varying cohorts of animals were euthanized at varying time points with some animals being euthanized after 6-8 weeks of age following initial surgery.

The numbers of animals included to generate each data set has been clarified in each figure legend.

- Why was no HI+saline ip group included?

Authors Response:

We did not include animals with HI + saline (ip) since 200�l of NS injected IP is not informative. Given the slow absorption of normal saline from the intraperitoneal compartment into blood an additional circulating volume of 10 cc/kg is not expected to modify the evolution of HIE. This would be a wastage of animals with limited useful data being generated. The neuroprotective effect of hypothermia is well established (1) which we have included as a comparative arm.

- Line 114 says Pentobarbital was used, line 134-140 does not include pentobarbital. Please explain.

Authors Response:

Euthanasia was routinely performed using a lethal dose of pentobarbital for all experiments performed.

Statistics:

- Why was mean and SEM used and not median and IQR? Was the data normally distributed? Can the authors show dot blots with the result of each individual animal, so the reader can see the distribution of inury?

Authors Response:

Mean and SEM are standard measurement used for animal experiments. In experiments utilizing rats of a single strain that have been in bred, housed under the same conditions, eating the same diet and undergoing identical procedures, the variability is highly constrained. In the analysis of human data, median and IQR measurements would be appropriate due to the high degree of variability associated with varying genetic backgrounds, age, diet, lifestyle, etc. Non-parametric tests are warranted for small sample sizes where there is reason to doubt that the data conforms with a normal distribution. Our data show limited evidence of skew and we have no reason to suspect that the data is not normally distributed.

Dot blots have been shown for the neurobehavioral experiments (figures 4 and 5) with the largest variability depicting the distribution of data.

Results:

- For S1A/B only n=3 or 4 animals were used. This seems very low number with the known large variability in this model. Can the authors please show dot blots? Do you have coronal MRI slices showing the infarction?

Authors Response:

Figure S1 A and B were pilot studies conducted to determine optimal dosing based on gross examination of the brain with the animals subjected to hypoxic ischemic insult without further receipt of therapeutic interventions including hypothermia. Dot blots are not available since the degree of injury was not quantified. Instead, we quantified the percentage of animals (shown in S1B) with injury , representative gross images of which has been shown for reference.

This animal model with similar number of animals (n=3-5) has been published prior with reliable results (1, 4). This practice is also in keeping with the ethical consideration against conducting animal experiments with large numbers of animals when much smaller numbers (e.g. n = 4) can yield meaningful results. Please see the following reference detailing the argument for utilizing the minimal numbers of animals possible and t-tests in animal experimentation (Michael F. W. Festing, Douglas G. Altman, Guidelines for the Design and Statistical Analysis of Experiments Using Laboratory Animals, ILAR Journal, Volume 43, Issue 4, 2002, Pages 244–258 https://doi.org/10.1093/ilar.43.4.244).

Representative coronal MRI images of T2 map taken at 24 hrs of life have been included as a supplemental figure 3 comparing normothermic animals with animals who received HT+ RLS-0071 which confirms the developed lesion in NT animals. 

- Figure 1: did the authors also analyse different brain areas, e.g. hippocampus? If not, why not? Where in the cortex was the analysis performed, how many fields of view? Again, number of animals appears low, please explain.

Author’s response: 

For the CV staining both hippocampus and the cortex were analyzed but only the data form the cortex used to generate Fig 2 panel D. The entire cortex was analyzed as 1 view with assurance of non-overlapping between images for all the animals. The cortex was specifically chosen to reduce sampling error which would be a concern when using multiple readings of the hippocampus given its small area. This is also supported by similar methods used in prior publications (1, 4) where in for technical feasibility reasons the consistently visualized cortex was selected for measurement of injury induced to the cortex. 

In a neonatal piglet model of HIE, the brain regions that are selectively vulnerable to hypoxia–ischemia (HI) correspond to those in human term newborns and HI causes cortical laminar necrosis in sensorimotor cortex, and hypothermia has been shown to decrease ischemic neuronal necrosis in this model (7). 

For additional explanation of low animals numbers please refer to prior response. 

- Why was HT not neuroprotective in this model? Maybe too little number of animals?

Author’s response: 

Our data shows that HT is neuroprotective and that RLS-0071 serves to augment the neuroprotection offered by HT. RLS-0071 improves on the cortical neuronal protection offered by HT as shown on histological evaluation. Further evidence of this rescue is best seen on neuroimaging studies where HT+ RLS-0071 reduces the volume of cerebral lesion induced at 24hrs. The images from day 21 show improved brain growth of the unaffected side at day 21 when receipt of neuroprotective therapies including HT and RLS-0071. 

- Figure 3: is is the large variability due to the small sample sizes and model variability?

Author’s response: 

We agree with the reviewer regarding the large variability making Fig 3 panel B difficult to interpret; thus we have elected to remove Figure 3 panel B. The remainder is now a supplemental figure with errors of margin expected with some variability given the small sample size and model variability.

- How many animals had MRI perfomed? Was it the same animal at day 1 and 21?

Author’s response: 

For the neuroradiological studies, 10 animals were included in each group and MRI acquisitions were performed for each animal at 24hr and 21days after the hypoxia/ ischemic injury.

- Why was hypothermia not neuroprotective measured by long term outcome? This is very surprising given the known literature.

Author’s response: 

Thank you for raising the point. As described above, our data shows that RLS-0071 augments the neuroprotective benefit hypothermia offers currently. 

Prior to the cooling era approximately 26.4% of infants with HIE survived with moderate to severe neurodevelopmental impairment and a further 14% survived with mild impairment. Reported rates of cerebral palsy following HIE are generally around 10%-13% among survivors of moderate to severe encephalopathy (8, 9). Sensory disruption is also increased following hypoxic-ischemic injuries. 

Although therapeutic hypothermia has improved the outlook for infants with moderate to severe HIE, with increased likelihood of survival with normal IQ and improved survival without neurological abnormalities, it is important to note that learning deficits may be present with or without motor or sensory dysfunction. Impairments in episodic memory associated with reduced hippocampal volume has been found in children following perinatal hypoxic-ischemic injury but without associated neurological deficits (10). Marlow et al (11) demonstrated memory and attention/executive function impairments in the severe encephalopathy group .

We demonstrate that HT+ RLS-0071 offers improvement in memory (ability to remember familiar objects) and long-term spatial memory retention both contributors towards enhanced learning.

Discussion:

- Please comment on the role of RLS-0071 as free radical scaveneger and the possibility of mechanism of action for the shown results in the study

Author’s response: 

Thank you for pointing out the additional mechanisms of action that could be at play. RLS-0071 has known antioxidant and neutrophil modulating properties exhibited by inhibition of NETosis (12). Given the role that oxidative stress contributes towards brain injury along with increasing evidence demonstrating the role of NETs in reperfusion related inflammatory brain damage, we speculate that in addition to classical complement inhibition, RLS-0071 can provide additional neuroprotection over hypothermia alone through alternative mechanisms. 

- The discussion is too short and superficial and should be improved.

Author’s response: Thank you for the feedback. We have improved on the discussion.

Minor:

- Line 28: HIE instead of HIA

Author’s response: 

Thank you for pointing out the oversight. We have corrected this.

Reviewer #3: Reviewer

In the study PONE-D-21-14373, Kumar and colleagues tested the effects a classical complement pathway inhibitor (RLS-0071) associated, or not, with hypothermia in the Vannucci model of neonatal HI in Wistar rats. Authors state that adjunctive therapy targeting inhibiting complement system improves the benefits afforded by therapeutical hypothermia. In general, the study has a clear hypothesis, the references used to sustain the results are adequate, however the study needs improvements in the description of the methods, results and a real discussion of the findings (which is not present in this version) and clear conclusion message. Despite the good level of grammar, some sentences need to be reorganized in order to make more understandable for the readers.

General Comments

# The legends of the Figures should be improved and moved to the end of the manuscript, since it is difficult to follow the sequence of the manuscript.

Author’s response: 

The figure legends for the main figures have been included to the manuscript body after the first mention of the figure in keeping with specifications from the journal. The figure legends of the supplemental figures are included at the end of the manuscript.

# The general scheme of administration, and especially the RLS-0071 in the HT groups should be better explained in the methods section.

Author’s response: 

Thank you for the suggestion. This has been clarified.

# The Discussion section need to be re-written and discuss in deep the data obtained in the study.

Author’s response: Thank you for the feedback. We have improved on the discussion.

Abstract and Intro

# Line 28 - The short name for Hypoxia-ischemia should be HIE.

Author’s response: 

Thank you for the suggestion. Hypoxia-ischemia has been referred to as H/I and hypoxic ischemic encephalopathy has been referred to as HIE.

# HI animals received 4 interventions? “one of the 4” is more adequate.

Author’s response: Thank you for the suggestion. This has been incorporated. 

# Please consider using the same terminology (RLS-0071) or PIC1 in the whole manuscript;

Author’s response: Thank you for the suggestion. This has been incorporated.

Methods

# In the session materials (l 91 to 94), please describe the code of antibodies used.

Author’s response: 

Where available the catalog numbers of the antibodies used has been added to the manuscript. 

# The authors perform the surgery from P10 to P12. However, due to the nature of the lesion, as well as brain energetic metabolism differences that can emerge from this range, do the authors have a control for the variability of the lesion? Despite the observed variability observed in the biochemical analysis, it is possible to observe it in the Rota-Rod test (Figure 6B - NT group).

Author’s response: 

We have clarified that the surgery was performed on P10 animals. To limit variability, all of the hypoxia/ischemia injury inducing surgeries at the primary lab (i.e results generated pertaining to gross examination, histology and cranial C1q ) were performed in Wistar rat pups by the same individual who was trained in the procedure by Susan Vannucci and through the course of these experiments performed nearly 200 procedures in the same lab set up with the same instruments in under 8 minutes with high reliability and on target performance. For the neuroradiological studies an additional Naïve group of animals were utilized to serve as a control. The extensive amount of work done on addressing variability has been described by Shah et al (1, 4).

# It is important to mention in the section Animal model, the hemisphere used in the study and the sex of the animals.

Author’s response: 

Thank you for the suggestion. We have clarified this in the manuscript. The right hemisphere was used and both male and female pups were included in all the studies.

# Were the sentinel animals included in the study? Some authors highly recommend to remove these animals from the analysis due to the stress caused by the rectal monitoring for this prolonged period.

Author’s response: 

Yes, sentinel animals were included in the study. Since we did not see any major outliers in the variability among the sentinel animals, we elected to incorporate them in to the result generated given the small numbers of animals (<5). All the animal experiments were performed under an approved protocol by the Eastern Virginia Medical School (EVMS) Institutional Animal Care and Use Committee (IACUC) and where applicable, all animal experiments were carried out according to the National Institute of Health (NIH) guidelines for the care and use of laboratory animals and approved by the National Ethics Committee (National Animal Experiment Board, Finland).

# How was the rewarming procedure? Did it take the 60 minutes recovery? Was the return to normothermia constant? Please, describe in more details.

Authors response: 

A prior established methodology (1) for rewarming and recovery was used. This was in keeping with IACUC recommendations. The animals were recovered on a homeothermic blanket system while maintaining normothermia at 37.0 ± 1 °C. The animals used varying times to recover but all of them recovered in 60 minutes. When all the animals had recovered , they were returned to the nursing dam en masse. 

# The description of sham group (l. 130) is not correct according what is a sham group (all procedures, without injury). It should be called as naïve, if there is no manipulation (line 131 and 132).

Authors response: 

Thank you for the suggestion. Naïve group has been clarified as group of animals who did not undergo any surgical procedure and were kept at normothermia for 6 hours. 

# Do the authors consider using only “sham” animals as the controls for the lesion in the MRI study, since a large number of studies” state that the contralateral hemisphere is not exactly a control? Is it possible to use Sham+normo as the control group?

Authors response:

We have included a supplement figure (S4 fig) which demonstrates consistently similar distribution of injury to the control hemisphere (contralateral) with no changes seen in T2 value between study groups (pooled genders). 

# For the neuroimaging studies animals were injected subcutaneously, what is the reason?

Authors response:

The neuroimaging studies were done by a commercial vendor, Charles River Discovery Research Services Finland Ltd. RLS-0071 was provided as ‘ready to use’ formulations to the vendor and the drug was administered as a fixed volume of administration at 200�l subcutaneous per institutional policy. The route of administration was switched from intraperitoneal to subcutaneous administration due to the veterinarian policy at the commercial vendor site.

# Please add the references for used in the Neurobehavioral tests.

Authors response:

Pertinent references have been added to the manuscript

# The difference in the number of animals used for the behavioral analysis has an explanation? Was there a cut-off for the Rota-Rod test?

Authors response:

The rotarod task is capable of detecting statistically significant injury induced motor impairment at a lower level of injury than the beam-balance or beam-walking tasks. At the moderate level of injury, the rotarod yielded highly significant differences between injured and uninjured animals. Rotard task is a more sensitive index of injury induced motor impairment than either the beam-balance or beam-walking latency. Additionally, rotarod task is a much more powerful procedure for testing motor dysfunction meaning that fewer subjects are required to detect injury induced motor impairment and to test the therapeutic effectiveness of pharmacologic interventions with the rotarod task. Hamm et al demonstrated that the rotarod task required a substantially smaller sample size, by a factor of at least 3, than that required by both the beam-balance and beam-walking tasks to reach the same degree of statistical power. Thus only a group of animals were included in the rotarod testing given the sensitive and efficient nature of the measure (13). As part of the study done by the commercial vendor, Charles River Discovery Research Services Finland Ltd. we also conducted a beam balance test to measure front paw and hind paw slips. We have not included this data given the similar results and the increase sensitivity of the rotarod task at measuring motor impairment. 

# Regarding the Novel Object Recognition test, authors performed the test in the same day of second training session, or next day? If it is 24h interval period, authors should consider long-term instead short-term memory evaluation.

Authors response: We agree with the reviewer and have modified it as an evaluation of long-term memory evaluation. The testing was performed to evaluate the influence of RLS-0071 on the animal’s memory and recognition as a sensitive measure of altered animal behavior (14). 

The following has been added to the manuscript:

Novel object recognition was performed to evaluate the influence of RLS-0071 on the animal’s memory and recognition as a sensitive measure of altered animal behavior (14). It serves as a test for long term object memory performance and is a sensitive measurement available for evaluating cognition enhancing activity of compounds (15).

Results

# Figure 1 - Please add a scale bar.

Authors response:

Thank you for the suggestion. This has been modified in the figure. 

# How homogeneous was the injury procedure? Was the number of animals (Fig S1) enough to reach this initial assumption of tissue protection? No statistical analysis was performed? Could you use an accepted classification of injury? The one proposed by by Palmer 1992 (003 1-3998/93/3304-0405S03.00/0)?

Authors response:

The procedure was conducted with minimal variability in performance and setup as described previously. 

Fig S1 was performed as a follow up to a prior a pilot study where in rescue doses of RLS-0071 varied from 160 mg/kg x2 down to 10 mg/kg x2. In those experiments, all dosing groups showed decreased numbers of brains with gross infarction, suggesting that RLS-0071 given one hour after infarction could decrease the numbers of animals experiencing significant infarction. However, all RLS-0071 dosage groups showed similar percentages of animals without gross infarction, suggesting that a minimal effective dose had not yet been achieved. No statistical analysis were performed on the data collected which showed percentage of animals with obvious damage visible on gross inspection. The overarching goal was to perform a dose range finding study and determine the minimal effective dose to carry forward RLS-0071 into drug development.

# Did author have an explanation for such variability in C1q levels in the brain at 8h? it seems from the hemolysis evaluation (Figure S1 ) that the variability of the inhibitor is very small until 8h. The variability of the model explains this??

Authors response:

Yes, we agree the model variability could explain the variability of the C1q levels in the brain measured at 8 hrs. Supplemental figure 1 has been removed 

# Figure 3a - Please refer, to Brain C1q levels, instead to Cranial.

Authors response:

Thank you for the suggestion. This has been modified in the figure. 

# The neuroimaging results, in which brain “growth” was stated, could be re-written. Normal brain volume, perhaps?

Authors response:

Thank you for the suggestion. We have edited the statement to reflect how HT+ RLS-0071 decreased cerebral injury in animals subjected to H/I insult allowing for continued brain growth following the insult.

# Line 413…. Despite no statistical difference (because it is 0.08), a trend to decreased latency….

Authors response:

Thank you for the suggestion.

# Is there a difference among the 4 groups in NOR regarding the exploration of familiar object?

Authors response:

There was no difference among the 4 groups in regard to the exploration of familiar objects.

# the description of Rota rod as an ability test, is very imprecise. RR tests animals’ coordination, balance, learning but not agility…Please, review. Despite this, the results (3 animals in NT and 5 in HI) clearly show the variability of the model, which could bias the final conclusions of the study.

Authors response: 

Thank you for the suggestion. We have edited the description of Rota rod as an indicator of sensorimotor coordination and added the following to the manuscript:

Rotarod task has been shown to be a sensitive and efficient measure of motor dysfunction after brain injury (13) 

Despite the very promising results from the group, based in the data here presented, I could not recommend the manuscript for publication in the present format.

6. PLOS authors have the option to publish the peer review history of their article (what does this mean?). If published, this will include your full peer review and any attached files.

Do you want your identity to be public for this peer review? For information about this choice, including consent withdrawal, please see our Privacy Policy.

Reviewer #1: No

Reviewer #2: No

Reviewer #3: Yes: Eduardo Farias Sanches

1. Shah TA, Pallera HK, Kaszowski CL, Bass WT, Lattanzio FA. Therapeutic Hypothermia Inhibits the Classical Complement Pathway in a Rat Model of Neonatal Hypoxic-Ischemic Encephalopathy. Frontiers in Neuroscience. 2021;15(114).

2. Patel SD, Pierce L, Ciardiello A, Hutton A, Paskewitz S, Aronowitz E, et al. Therapeutic hypothermia and hypoxia–ischemia in the term-equivalent neonatal rat: characterization of a translational preclinical model. Pediatric research. 2015;78(3):264-71.

3. Vannucci SJ, Seaman LB, Vannucci RC. Effects of hypoxia-ischemia on GLUT1 and GLUT3 glucose transporters in immature rat brain. Journal of Cerebral Blood Flow & Metabolism. 1996;16(1):77-81.

4. Shah TA, Nejad JE, Pallera HK, Lattanzio FA, Farhat R, Kumar PS, et al. Therapeutic hypothermia modulates complement factor C3a and C5a levels in a rat model of hypoxic ischemic encephalopathy. Pediatric research. 2017;81(4):654-62.

5. Sun H, Juul HM, Jensen FE. Models of hypoxia and ischemia-induced seizures. J Neurosci Methods. 2016;260:252-60.

6. Hamdy N, Eide S, Sun HS, Feng ZP. Animal models for neonatal brain injury induced by hypoxic ischemic conditions in rodents. Exp Neurol. 2020;334:113457.

7. Wang B, Armstrong JS, Lee J-H, Bhalala U, Kulikowicz E, Zhang H, et al. Rewarming from Therapeutic Hypothermia Induces Cortical Neuron Apoptosis in a Swine Model of Neonatal Hypoxic–Ischemic Encephalopathy. Journal of Cerebral Blood Flow & Metabolism. 2015;35(5):781-93.

8. Rennie JM, Hagmann CF, Robertson NJ, editors. Outcome after intrapartum hypoxic ischaemia at term. Seminars in Fetal and Neonatal Medicine; 2007: Elsevier.

9. Dixon G, Badawi N, Kurinczuk JJ, Keogh JM, Silburn SR, Zubrick SR, et al. Early developmental outcomes after newborn encephalopathy. Pediatrics. 2002;109(1):26-33.

10. Gadian DG, Aicardi J, Watkins KE, Porter DA, Mishkin M, Vargha-Khadem F. Developmental amnesia associated with early hypoxic–ischaemic injury. Brain. 2000;123(3):499-507.

11. Marlow N, Rose A, Rands C, Draper E. Neuropsychological and educational problems at school age associated with neonatal encephalopathy. Archives of Disease in Childhood-Fetal and Neonatal Edition. 2005;90(5):F380-F7.

12. Hair PS, Enos AI, Krishna NK, Cunnion KM. Inhibition of complement activation, myeloperoxidase, NET formation and oxidant activity by PIC1 peptide variants. PLoS One. 2019;14(12):e0226875.

13. Hamm RJ, Pike BR, O'DELL DM, Lyeth BG, Jenkins LW. The rotarod test: an evaluation of its effectiveness in assessing motor deficits following traumatic brain injury. Journal of neurotrauma. 1994;11(2):187-96.

14. Antunes M, Biala G. The novel object recognition memory: neurobiology, test procedure, and its modifications. Cogn Process. 2012;13(2):93-110.

15. Mathiasen JR, DiCamillo A. Novel object recognition in the rat: a facile assay for cognitive function. Current protocols in pharmacology. 2010;49(1):5.59. 1-5.. 15.

---

## [Decision Letter · Decision Letter 1]

15 Sep 2021

Classical Complement Pathway Inhibition Reduces Brain Damage in a Hypoxic Ischemic Encephalopathy Animal Model

PONE-D-21-14373R1

Dear Dr. Kumar,

We’re pleased to inform you that your manuscript has been judged scientifically suitable for publication and will be formally accepted for publication once it meets all outstanding technical requirements.

Kind regards,

Olivier Baud, MD, PhD

Academic Editor

PLOS ONE

Additional Editor Comments (optional):

Reviewers' comments:

Reviewer's Responses to Questions

**Comments to the Author**

1. If the authors have adequately addressed your comments raised in a previous round of review and you feel that this manuscript is now acceptable for publication, you may indicate that here to bypass the “Comments to the Author” section, enter your conflict of interest statement in the “Confidential to Editor” section, and submit your "Accept" recommendation.

Reviewer #2: All comments have been addressed

2. Is the manuscript technically sound, and do the data support the conclusions?

Reviewer #2: Yes

3. Has the statistical analysis been performed appropriately and rigorously? 

Reviewer #2: Yes

4. Have the authors made all data underlying the findings in their manuscript fully available?

Reviewer #2: Yes

5. Is the manuscript presented in an intelligible fashion and written in standard English?

Reviewer #2: Yes

6. Review Comments to the Author

Reviewer #2: The authors have done a great job revising the paper. It is much easier to read and follow the experiments. I have no further comments or questions.

7. PLOS authors have the option to publish the peer review history of their article (what does this mean?). If published, this will include your full peer review and any attached files.

Reviewer #2: No

---

## [Editor Report · Acceptance letter]

23 Sep 2021

PONE-D-21-14373R1 

Classical Complement Pathway Inhibition Reduces Brain Damage in a Hypoxic Ischemic Encephalopathy Animal Model 

Dear Dr. Kumar:

I'm pleased to inform you that your manuscript has been deemed suitable for publication in PLOS ONE. Congratulations! Your manuscript is now with our production department. 

Kind regards, 

on behalf of

Pr. Olivier Baud 

Academic Editor

PLOS ONE